# The Influence of Frailty Syndrome and Dementia on the Convenience and Satisfaction with Oral Anticoagulation Treatment in Elderly Patients with Atrial Fibrillation

**DOI:** 10.3390/ijerph19095355

**Published:** 2022-04-28

**Authors:** Katarzyna Lomper, Maria Łoboz-Rudnicka, Tomasz Bańkowski, Krystyna Łoboz-Grudzień, Joanna Jaroch

**Affiliations:** 1Department of Clinical Nursing, Medical University, 51-618 Wroclaw, Poland; j.jaroch@wp.pl; 2Department of Cardiology, T. Marciniak Hospital, 54-049 Wroclaw, Poland; marialoboz@o2.pl (M.Ł.-R.); tba0@poczta.onet.pl (T.B.); kloboz@wp.pl (K.Ł.-G.)

**Keywords:** atrial fibrillation, oral anticoagulation, treatment satisfaction, treatment convenience, geriatric elements

## Abstract

Background: The impact of frailty syndrome (FS) and dementia on the convenience and satisfaction with oral anticoagulation (OAC) treatment in atrial fibrillation (AF) patients is not well-known. Aim: Assessment the impact of FS and dementia on the convenience and satisfaction with OAC treatment in 116 elderly (mean age 75.2, SD = 8.2) patients with AF. Methodology: A self-administered questionnaire was used in the study to collect basic socio-demographic and clinical data. Tilburg Frailty Indicator (TFI) questionnaire was used to assess the presence of FS, Mini Mental State Examination (MMSE) to assess cognitive impairment (CI), The Perception of Anticoagulant Treatment Questionnaire Part 2 (PACT-Q2) to assess convenience and satisfaction with OAC treatment, and the Arrhythmia-Specific Questionnaire in Tachycardia and Arrhythmia (ASTA) to assess quality of life (QoL). Results: Multivariable analysis as a significant, negative predictor of the convenience and satisfaction domain showed the occurrence of dementia (β = −0.34; *p* < 0.001, β = −0.41; *p* < 0.001, respectively) and prior major bleeding (β = −0.30; *p* < 0.001, β = −0.33; *p* < 0.001, respectively). Analysis showed a significant relationship between convenience and satisfaction and the overall result of the ASTA (r = −0.329; *p* < 0.001, r = −0.372; *p* < 0.001, respectively). Conclusions: Elements of geriatric syndrome, such as FS and dementia, adversely affect treatment convenience and satisfaction with OAC treatment in AF. It has been shown that better convenience and satisfaction with OAC treatment translates into better QoL. There were no differences between satisfaction and convenience and the type of OAC treatment (vitamin K antagonists (VKA) vs. novel oral anticoagulants (NOAC).

## 1. Introduction

Decisions regarding oral anticaogulation (OAC) treatment in older adults with atrial fibrillation (AF) require a holistic aproach. Comprehensive geriatric assessment (CGA), including frailty syndrome (FS), cognitive impairment (CI), and bleeding risk, may be important in clinical decisions in patients with AF. FS and CI are both conditions associated with risk of mortality, but the presence of FS and CI does not appear to be contraindication to anticoagulation in AF. There is a scant evidence of net clinical benefit in patients with frailty and cognitive functional impairment as a surrogate marker of biological age [1].

In the literature, more atention is being given to patient-reported outcomes (PROs). PROs provide important information from the patient’s perspective and the subjective measures relating to patent experience and quantify assessment of patient satisfaction, adherence, or quality of life (QoL) [2]. Antigoagulation convienience and satisfaction are an important PROs markers. The link between phisical, cognitive conditions with anticoagulation convienience and satisfaction are not well-established. Higher values of convenience and satisfaction during novel oral anticoagulants (NOAC) treatment should be expected compared to the vitamin K antagonists (VKA).

Better convienience and satisfaction with OAC treatment is important, as it translates into better adherence and QoL [3]. There are single studies investigating the effect of FS and cognitive impairment on antithrombotic treatment satisfaction although they do not assess the effect of dementia [4]. There is also a lack of papers evaluating the effect of FS and dementia on the convenience of OAC treatment in this group of patients.

### Aim of the Study

We aimed to assess the impact of FS and dementia on treatment convenience and satisfaction with OAC in patients with AF. The study also aims to examine how the type of OAC treatment, namely VKA or NOAC, affects convenience and satisfaction and assess the impact of convenience and satisfaction on QoL.

## 2. Material and Methods

### 2.1. Study Settings and Patrticipants

The study was carried out on a group of 116 elderly (mean age 75.2, SD = 8.2, 64 women and 52 men), randomly selected in-patients with non-valvular AF hospitalized in the Department of Cardiology in the T. Marciniak Lower Silesian Specialist Hospital in Wroclaw. The study was cross-sectional and questionnaire-based. The inclusion criteria were: non-valvular atrial fibrillation, age over 60, taking oral anticoagulation therapy for at least 6 months, and providing written and informed consent to participate in the study. The exclusion criteria were: condition of the patient requiring intensive cardiac care, lack of written and informed consent to participate in the study, mental disorders, and a history of stroke within the last 6 months. All patients qualified to participate in the study were asked to give their written consent to the study and to fill out the study questionnaires in the presence of the investigator. In our study, 19 patients had a history of stroke more than 6 months before participation in the survey.

### 2.2. Research Tools

Basic sociodemographic and clinical data were obtained using the author’s questionnaire and analysis of medical records. In the study, the severity of disease symptoms according to the European Heart Rhythm Association was used. The CHA_2_DS_2_VASc scale was used to determine the risk of thromboembolism. The study also used the HAS-BLED bleeding risk assessment scale [5].

The assessment of the occurrence of frailty syndrome was based on the Tilburg Frailty Indicator (TFI) questionnaire in the Polish-language version. The TFI was developed on the basis of an integral model of frailty. The scale consists of two parts: part A, concerning health determinants of FS, and part B, concerning 15 questions on the occurrence of the main components of frailty. The total result of TFI can range from 0 to 15 points, and FS is recognized at 5 points and above [6,7].

The Mini Mental State Examination (MMSE) questionnaire was used to determine cognitive function. The total score of the questionnaire may range from 0 to 30 points, with lower scores indicating a cognitive impairment and dementia. A score of 30–27 points means no cognitive dysfunction, while a score of 26–24 points is equivalent to cognitive impairment without dementia. Dementia is recognized by a score of 23 points or fewer [8,9].

To assess the quality of life, a standardized research tool was used: the Arrhythmia-Specific Questionnaire in Tachycardia and Arrhythmia (ASTA), part III, questionnaire in the Polish-language version. Part III of the questionnaire was used in the study to assess the influence of arrhythmia on the patients’ daily life and HRQoL. The ASTA HRQoL total scale score ranges from 0 (best possible HRQoL) to 39 (worst possible HRQoL). Higher scores reflect a more negative impact of arrhythmia on HRQoL [10,11,12].

The evaluation of antithrombotic treatment satisfaction and convenience was performed by using the Perception of Anticoagulant Treatment Questionnaire Part 2. (PACT-Q2). The PACT-Q2 measures convenience and satisfaction and is administered to patients once treatment is ongoing. The total score for each domain ranges from 0 to 100, with higher values corresponding to higher level of convenience and satisfaction with OAC treatment [13,14].

### 2.3. Ethical Considerations

The study was approved by the independent Bioethics Committee of the Wroclaw Medical University, Poland. The study was carried out in accordance with the tenets of the Declaration of Helsinki and guidelines of good clinical practice.

### 2.4. Statistical Analysis

The statistical analysis was performed using Statistica 9.0 EN. The level of statistical significance was considered *p* < 0.05. Mann–Whitney test was used to compare quantitative variables between two groups, while Kruskal–Wallis test (followed by Dunn post hoc test) was used for more than two groups. The relationship between two quantitative variables was assessed with Spearman’s coefficient of correlation. Uni- and multivariable linear regressions were used to analyze impact of potential predictors on quantitative variables.

## 3. Results

The average age of the studied group was 75.2 years (SD = 8.2). More than half of the respondents (56%) were patients with diagnosed permanent AF. A total of 66.4% of patients received VKA treatment. FS was diagnosed in 67.2% and dementia in 36.2% of patients. Basic data are presented in Table 1.


The analysis of the PACT-Q2 questionnaire showed an average score of the convenience domain at 81.3 points, while the satisfaction domain at 58.7 points.

Based on comparative analysis, lower domain of convenience values were observed in the group with diagnosed FS (mean ± SD 78.5 ± 14.3) compared to those with no FS (mean ± SD 88.3 ± 11.8) and with dementia (mean ± SD 73.8 ± 14.8) as compared to those with cognitive impairment without dementia (mean ± SD 88 ± 9.2) and with normal cognitive function (mean ± SD 84.4 ± 13.8).

The lower satisfaction values were reported in the group of patients with diagnosed FS (mean ± SD 55.5 ± 13.3) than in group with no FS (mean ± SD 65 ± 11.1) and in group with dementia (mean ± SD 49.9 ± 11) than in groups with cognitive impairment without dementia and with normal cognitive function (mean ± SD 61.4 ± 11 and 65.8 ± 12.7, respectively). The data are presented in Table 2.

In a single-variable analysis, FS (β = −0.32; *p* < 0.001), dementia (β = −0.42; *p* < 0.001) and prior major bleeding (β = −0.30; *p* < 0.001) were shown to have a negative effect on the convenience domain. There was a significant, positive effect of professional activity (β = 0.29; *p* < 0.001) and self-administration of medication (β = 0.27; *p* < 0.001) on the convenience domain (Table 3).

Multi-variable analysis as a significant, negative predictor of the convenience showed the occurrence of dementia (β = −0.34; *p* < 0.001) and prior major bleeding (β = −0.30; *p* < 0.001). As a significant, positive predictor, we observed professional activity (β = 0.17; *p* < 0.05) and self-administration of medicines (β = 0.29; *p* < 0.001). The data are presented in Table 3.

In the single-variable analysis, we observed a negative effect of EHRA classification III and IV (β = −0.3; *p* < 0.001), disease duration > 5 years (β = −0.23; *p* = 0.01), FS (β = −0.33; *p* < 0.001), dementia (β = −0.5; *p* < 0.001), and prior major bleeding (β = −0.34; *p* < 0.001) on treatment satisfaction. As a significant, positive predictor, we observed professional activity (β = 0.23; *p* = 0.01), self-administration of medicines (β = 0.24; *p* = 0.007), and being in a relationship (β = 0.24; *p* = 0.01).

In multi-variable analysis, prior major bleeding (β = −0.33; *p* < 0.001) and the presence of dementia (β = −0.41; *p* < 0.001) were a significant negative predictor of treatment satisfaction. As a significant, positive predictor, we observed self-administration of medicines (β = 0.19; *p* = 0.02). The data are presented in Table 4.

In ASTA questionnaire higher scores reflect a more negative impact of arrhythmia on QoL—the fewer the points, the better QoL. In the PACT-Q2 higher values corresponding to higher level of convenience and satisfaction with OAC treatment. Analysis showed a significant relationship between convenience and satisfaction and the overall result of the ASTA (r = −0.329; *p* < 0.001, r = −0.372; *p* < 0.001, respectively) and its physical domain (r = −0.356; *p* < 0.001, r = −0.374; *p* < 0.001, respectively). There was also a significant correlation between the PACT-Q2 satisfaction domain and mental domain of ASTA (r = −0.303; *p* < 0.001). The data are presented in Table 5.

## 4. Discussion

There are few studies assessing satisfaction of OAC treatment in the elderly in AF [3,4] or both satisfaction and convenience [15]. Convenience and satisfaction with OAC treatment is an important factor in assessing the effectiveness of treatment. Patients who are satisfied with OAC treatment show better clinical parameters, such as the International Normalized Ratio (INR), and better QoL [3].

Age is associated with the risk of frailty syndrome (FS) and dementia. Both FS and CI are indicated as one of the following for the most frequently mentioned reasons of lack of adherence [16]. The incidence of FS in patients with atrial fibrillation is estimated between 4.4% to 75.4% [17], and 67.2% of patients reported FS in our study. FS is one of the risk factors for earlier development of cognitive impairment [18].

It is still unknown whether antithrombotic treatment delays the development of CI. The relationship between AF and CI, including dementia, has been confirmed by meta-analyses. Kalantarian et al., in a meta-analysis of 89.907 participants, showed that AF is associated with a 40% increase in the risk of CI regardless of the stroke [19]. Wozakowska-Kaplon et al. demonstrated that the permanent arrhythmia among people over 65 years of age may be associated with lower results in the MMSE questionnaire, and CI was diagnosed in 43% of respondents [20]. In our study, CI without dementia occurred in 31.9% of respondents, while dementia occurred in 36.2%.

In our study, using the PACT-Q2 questionnaire, the average score for convenience of OACs was 81.3 ± 13.4 points, and the satisfaction score was 58.7 ±13.3 points. Brüggenjürgen et al. analyzed 5049 patients using the PACT-Q2 questionnaire. A similar mean score was shown for both convenience (82.9 ± 17.3) and satisfaction (63.4 ± 15.9 points [21]). Mull et al. showed a mean score of 87.88 (SD = 16.69) for convenience and a mean score of 67.86 (SD = 19.96) for satisfaction with OAC treatment in elderly patients with AF [15]. Gospos and Bernaitis showed that the mean overall score in AF patients for convenience was 72.26 ± 10.71 and 67.97 ± 18.58 for satisfaction [22].

A few studies in the literature evaluate the effect of FS and CI on OAC treatment satisfaction [4]. The SAGE-AF (Systematic Assessment of Geriatric Elements in Atrial Fibrillation) register on 1.444 patients > 65 years of age with AF who received OAC were included in the prospective study. The effect of six components of geriatric syndrome, namely FS, CI, social support, depression and anxiety, and hearing and vision disorders, on OAC treatment satisfaction measured by the Anticlot Treatment Scale (ACTS) was studied. The incidence of FS was significantly associated with lower satisfaction scores in univariate analysis [4]. Vision disorders, depression, and anxiety, all more common in elderly populations, had a negative impact on treatment satisfaction [4]. In a self-reported study, patients with FS and dementia scored worse in satisfaction and convenience of OAC treatment. In univariate analysis, frailty syndrome and dementia were significant negative predictors of satisfaction and convenience with OAC treatment although multivariate analysis showed the presence of dementia as a significant negative factor affecting both satisfaction and convenience.

The own study showed that the higher EHRA classification (EHRA III and EHRA IV) in the univariate analysis had a negative effect on satisfaction with OAC treatment. The severity of symptoms in AF can affect not only the patient’s daily life and function but also the patient’s sense of satisfaction with anticoagulation treatment.

Oral anticoagulation therapy is associated with an increased risk of bleeding. In our study, the history of bleeding determined the convenience and satisfaction results. Patients with a history of bleeding achieved lower mean scores for convenience (72.9 vs. 83.3 points) and satisfaction (49.7 vs. 60.9 points) compared to patients without previous bleeding. In the univariate and multivariable analysis, a history of bleeding was also a significant negative factor affecting treatment convenience and satisfaction. Weernink et al. showed that the presence of bleeding in the medical history is one of the main factors that determine patients’ perception of OAC therapy [23].

Additionally, in our study, self-administration of medicines without the support of family members were associated with higher values of PACTQ-2 questionnaire, and this was a significant positive predictor of convenience and satisfaction. Thus, self-administration of medications is an important factor affecting patients’ feelings while receiving oral anticoagulant therapy. Furthermore, in our study, disease duration > 5 years in univariate analysis had a negative effect on satisfaction with OAC treatment.

In clinical practice, VKA and NOAC are used in anticoagulation treatment. Although we expected higher values of convenience and satisfaction in the group of patients taking NOACs, in our study, we did not show statistically significant differences depending on the use of the OAC treatment (VKA or NOAC). However, patients taking NOAC received higher convenience values. Many studies have shown no differences in satisfaction with VKA treatment compared to NOAC. The multi-center SAFARI study, conducted on a group of 405 AF patients switching from VKA to NOAC (rivaroxaban) on the basis of the ACTS scale, showed improvement in treatment satisfaction after three months of treatment with rivaroxaban compared to the baseline condition (VKA) [24]. The RE-SONANCE (Patient Convenience study) register study showed better convenience and satisfaction assessed using the PACT-Q2 questionnaire in patients treated with dabigatran compared to the VKA [25].

OAC therapy can affect a patient’s subjective sense of QoL due to the need for lifestyle changes, dietary restrictions, and other limitations. The QoL of patients receiving OAC treatment is receiving greater attention in recent years. There are available studies that evaluate independently the QoL and satisfaction with OAC in the AF population [26], but there are few studies that indicate the effect of satisfaction with treatment on QoL among patients with diagnosed atrial fibrillation [27]. Our study showed a significant relationship between convenience and satisfaction with OAC treatment and quality of life. The available literature indicates that a higher level of satisfaction during OAC therapy is associated with better compliance and adherence to the therapeutic recommendations and therefore with the improvement of QoL [3,28].

Guidelines for the management of AF emphasizes the role of the patient in an integrated management approach to the therapeutic process. In the integrated treatment of AF, the need for multidisciplinary teams is indicated, the composition of which should depend on the individual needs, values, goals, and preferences of the patient [5]. Patients with FS and cognitive impairment, including dementia, would benefit from a multidisciplinary teams including not only specialists and AF nurses but also family members or caregivers.

## 5. Conclusions

Elements of geriatric syndrome, such as FS and dementia, adversely affect convenience and satisfaction with OAC treatment in AF. However, in our study, the only independent predictor of treatment satisfaction and convenience was dementia. According to the guidelines, FS is not a contraindication to starting OAC, so our study supports guidelines [5] (Section A.1).

Interestingly, among sociodemographic factors, we also noted a positive effect of professional activity on convenience and satisfaction with OAC treatment. There were no differences between satisfaction and convenience and the type of OAC treatment (VKA vs. NOAC). It has been shown that better convenience and satisfaction of OAC treatment translates into better QoL. However, the treatment convenience and satisfaction evaluation during OAC therapy in patients with AF with FS and dementia should be part of clinical practice. Understanding treatment convenience and satisfaction and its determinants can help to plan and succeed in OAC treatment.

## Figures and Tables

**Table 1 ijerph-19-05355-t001:** Sociodemographic and clinical characteristics of the study group (*n* = 116).

Average Age	75.2	SD = 8.2
Tested Parameter	*n*	% of the Total Group
**Sex**		
Women	64	55.2
**Marital status**		
In relationship	61	52.6
Single	55	47.4
**Professional activity**		
Professionally active	12	10.3
Retire	104	89.7
**Type of atrial fibrillation**		
Permanent	65	56.0
Paroxysmal/Persistent	51	44.0
**Duration of illness**		
Up to 5 years	69	59.5
Over 5 years	47	40.5
**EHRA Classification**		
EHRA I and II	57	49.1
EHRA III and IV	59	50.9
**Type of OAC treatment**		
VKA	77	66.4
NOAC	39	33.6
**The way of taking the medications**		
Self-administration	96	82.8
With help of family member/special pill box	20	17.2
**Comorbidities**		
3 and less	42	36.2
More than 3	74	63.8
**Frailty Syndrome (TFI)**		
Present	78	67.2
**Cognitive function assesment (MMSE)**		
No cognitive impairment	37	31.9
Cognitive impairment without dementia	37	31.9
Dementia	42	36.2
**Thromboembolic risk (CHA_2_DS_2_VASc)**		
Low (<2 pts)	2	1.7
High (≥2 pts in men, ≥3 pts in women)	114	98.3
**Bleeding risk (HAS-BLED)**		
Low (<3 pts)	58	50.0
High (≥3 pts)	58	50.0

Abbreviations: EHRA, The European Heart Rhythm Association; OAC, oral anticoagulation; NOAC, non-vitamin K antagonists; VKA, vitamin K antagonists; TFI, Tilburg Frailty Indicator; MMSE, Mini Mental State Examination; PTS, points.

**Table 2 ijerph-19-05355-t002:** Comparative analysis of sociodemographic and clinical variables for the PACT-Q2 convenience domain and for the PACT-Q2 satisfaction domain of the study group (*n* = 116).

Parameter	Group	Convenience				Satisfaction			
		Mean ± SD	Median	Quartiles	*p*	Mean ± SD	Median	Quartiles	*p*
**Age**	<64	80.3 ± 17.4	83.3	76–90.6	0.879	62.7 ± 15.4	67.9	58.9–71.4	0.1
	65–74	82.9 ± 12.8	85.4	77.6–91.1		61.1 ± 12.9	64.3	53.6–71.4	
	>74	81.1 ± 14.8	85.4	72.9–92.7		56.3 ± 13	53.6	46.4–67.9	
**Sex**	Female	81.4 ± 14.3	85.4	72.9–92.2	0.73	57.5 ± 13.2	57.1	46.4–67.9	0.367
	Male	82 ± 14.4	85.4	76.6–91.7		60 ± 13.5	64.3	50–71.4	
**Marital status**	In relationship	83.5 ± 13.1	85.4	79.2–93.8	0.141	61.9 ± 12.6	64.3	53.6–71.4	0.005 *
	Single	79.7 ± 15.3	79.2	70.8–91.7		55 ± 13.3	53.6	44.6–67.9	
**Professional activity**	Professionally active	94.1 ± 5.8	95.8	90.6–98.4	<0.001 *	67.6 ± 8	69.6	62.5–72.3	0.008 *
	Retired	80.3 ± 14.2	81.2	72.9–89.6		57.6 ± 13.5	57.1	46.4–67.9	
**Duration of illness**	Up to 5 years	84.4 ± 12.6	87.5	75–95.8	0.012 *	59.7 ± 12.3	60.7	50–67.9	0.316
	Over 5 years	77.7 ± 15.6	81.2	71.9–87.5		57.1 ± 14.8	53.6	46.4–69.6	
**Type of AF**	Paroxymal/Persistent	81.2 ± 14.8	85.4	74–92.7	0.905	57.8 ± 12.2	57.1	50–67.9	0.609
	Permanent	82.1 ± 13.9	83.3	75–91.7		59.3 ± 14.3	64.3	46.4–71.4	
**EHRA classification**	I and II	84.4 ± 14.6	89.6	79.2–93.8	0.006 *	62.8 ± 13.4	64.3	53.6–71.4	0.001 *
	III and IV	79.1 ± 13.5	81.2	70.8–87.5		54.7 ± 12.1	53.6	46.4–64.3	
**Bleeding risk (HAS-BLED)**	Low	82.5 ± 14.5	85.4	77.1–93.2	0.502	60.3 ± 14.1	64.3	46.4–71.4	0.154
	High	80.9 ± 14.1	81.2	72.9–91.7		57 ± 12.4	57.1	47.3–67.9	
**Thromboembolic risk (CHA_2_DS_2_VASc)**	Low	87.5 ± 12.8	92.7	84.4–95.8	0.347	66.1 ± 9.4	67.9	61.6–72.3	0.224
	High	81.5 ± 14.3	85.4	75–91.7		58.4 ± 13.4	58.9	46.4–67.9	
**Way of taking the medications**	Self-administration	83.5 ± 13.2	85.4	77.1–92.2	0.003 *	60.2 ± 13	64.3	50–71.4	0.013 *
	With help of family member/ special pill box	72.9 ± 16	75	66.1–82.3		51.4 ± 13.1	48.2	45.5–58	
**Type of OAC treatment**	VKA	80.1 ± 14.6	81.2	72.9–89.6	0.078	59 ± 13.8	64.3	46.4–71.4	0.649
	NOAC	84.8 ± 13.1	87.5	79.2–95.8		57.9 ± 12.6	57.1	46.4–67.9	
**Comorbidities**	Up to 3	83.3 ± 13.3	85.4	77.6–93.2	0.393	58.7 ± 12.6	64.3	47.3–67.9	0.963
	Over 3	80.8 ± 14.8	85.4	72.9–91.1		58.6 ± 13.8	57.1	46.4–71.4	
**Prior major bleeding** **or predisposition to bleeding**	No	83.8 ± 12.9	85.4	77.1–93.8	0.002 *	60.9 ± 12.7	64.3	50–71.4	<0.001 *
	Yes	73.1 ± 16.2	75	65.3–81.2		49.7 ± 12.1	50	42.9–57.1	
**Frailty symdrome (TFI)**	No	88.3 ± 11.8	89.6	85.4–95.8	<0.001 *	65 ± 11.1	67.9	61.6–71.4	<0.001 *
	Yes	78.5 ± 14.3	79.2	70.8–89.1		55.5 ± 13.3	53.6	46.4–64.3	
**Cognitive impairment (MMSE)**	Normal cognitive function	84.4 ± 13.8	87.5	81.2–95.8	<0.001 *	65.8 ± 12.7	67.9	64.3–71.4	<0.001 *
	Cognitive impairment without dementia	88 ± 9.2	89.6	79.2–95.8		61.4 ± 11	64.3	50–67.9	
	Cognitive impairment with dementia	73.8 ± 14.8	77.1	66.7–84.9		49.9 ± 11	48.2	42.9–56.2	

Abbreviations: AF, atrial fibrillation; OAC, oral anticoagulation; EHRA, The European Heart Rhythm Association; NOAC, non-vitamin K antagonists; VKA, vitamin K antagonists. Two groups comparison: *p*-Mann–Whitney test; >2 groups comparison: Kruskal–Wallis test + post hoc analysis (Dunn test) * statistically significant (*p* < 0.05).

**Table 3 ijerph-19-05355-t003:** Single- and multi-variable analysis PACT-Q2 convenience domain of the study group (*n* = 116).

Parameter	Single-Variable Jednoczynnikowa		Multi-Variable	
	Analysis		Analysis	
	β	*p*	β	*p*
EHRA classification III and IV	−0.18	0.052	0.05	0.57
Professionally active	**0.29**	**< 0.001**	**0.17**	**0.05**
Type of OAC treatment: VKA	−0.14	0.14	−0.09	0.29
Duration of AF > 5 years	0.03	0.73	−0.10	0.26
Self-administration of medicines	**0.27**	**<0.001**	**0.29**	**<0.001**
Female sex	−0.05	0.58	0.09	0.33
High thromboembolic risk (≥2 pcts men, ≥3 pcts women)	−0.11	0.24	0.00	0.98
High bleeding risk (HAS-BLED ≥ 3 pcts)	−0.05	0.62	0.16	0.07
Frailty syndrome	**−0.32**	**<0.001**	−0.04	0.65
Dementia	**−0.42**	**<0.001**	**−0.34**	**<0.001**
Marital status: in relationship	0.11	0.23	0.03	0.71
Prior major bleeding	**−0.30**	**<0.001**	**−0.30**	**<0.001**

Abbreviations: AF, atrial fibrillation; EHRA, The European Heart Rhythm Association; OAC, oral anticoagulation; VKA, vitamin K antagonists.

**Table 4 ijerph-19-05355-t004:** Single- and multi-variable analysis PACT-Q2 satisfaction domain of the study group (*n* = 116).

Parameter	Single-Variable		Multi-Variable	
	Analysis		Analysis	
	β	*p*	β	*p*
EHRA classification III and IV	**−0.3**	**<0.001**	−0.09	0.31
Professionally active	**0.23**	**0.01**	0.10	0.24
Type of OAC treatment: VKA	0.04	0.65	0.09	0.29
Duration of AF > 5 years	**−0.23**	**0.01**	0.03	0.75
Self-administration of medicines	**0.24**	**0.007**	**0.19**	**0.02**
Female sex	−0.09	0.32	0.10	0.24
High thromboembolic risk (≥2 pcts men, ≥3 pcts women)	−0.11	0.24	0.04	0.65
High vleeding risk (HAS-BLED ≥ 3 pcts)	−0.13	0.17	0.04	0.61
Frailty syndrome	**−0.33**	**<0.001**	−0.03	0.75
Dementia	**−0.5**	**<0.001**	**−0.41**	**<0.001**
Marital status: in relationship	**0.24**	**0.01**	0.13	0.16
Prior major bleeding	**−0.34**	**<0.001**	**−0.33**	**<0.001**

**Table 5 ijerph-19-05355-t005:** Analysis of correlation between convenience and satisfaction (PACT-Q2) and quality of life (ASTA) of the study group (*n* = 116).

Parameter	Spearman’s *Rank* *Correlation*		
	*n*	R Spearman	*p*
PACT-Q2 convenience and ASTA total score	116	−0.329	**<0.001**
PACT-Q2 convenience and ASTA physical domain	116	−0.356	**<0.001**
PACT-Q2 convenience and ASTA mental domain	116	−0.206	0.03
PACT-Q2 satisfaction and ASTA total score	116	−0.372	**<0.001**
PACT-Q2 satisfaction and ASTA physical domain	116	−0.374	**<0.001**
PACT-Q2 satisfaction and ASTA mental domain	116	−0.303	**<0.001**

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
