# Peer review of "The Influence of Frailty Syndrome and Dementia on the Convenience and Satisfaction with Oral Anticoagulation Treatment in Elderly Patients with Atrial Fibrillation"

_ijerph, 2022, doi:10.3390/ijerph19095355_

Round 1

Reviewer 1 Report

it is a well-done work and important for elder patients with AF. the author delare that dementia  adversely affect treatment convenience and satisfaction with OAC treatment in AF. however, is CI also affect them?

other factors in CGA, such as depression may also affect patients' satisfaction.

Author Response

Dear Editoriar Board,

Dear Reviewer,

We would like to thank the Editorial Board and the Reviewers for their feedback and constructive recommendations regarding our original research paper entitled:  “The influence of frailty syndrome and dementia on the convenience and satisfaction with oral anticoagulation treatment in elderly patients with atrial fibrillation.”

We would like to respond to this opinion based on our careful revision.

Sincerelly yours,

Katarzyna Lomper, corresponding author

Reviewer1

It is a well-done work and important for elder patients with AF. the author delare that dementia adversely affect treatment convenience and satisfaction with OAC treatment in AF. however, is CI also affect them? other factors in CGA, such as depression may also affect patients' satisfaction.

We want to thank the Reviewer 1 for insightful analysis and evaluation of our paper. We agree with the valid point that in addition to CGA, other factors such as depressive disorders may also affect patient satisfaction and convenience when receiving OAC therapy. In future papers, we will try to take the indicated factors into account. In our study, we found no significant difference in the comparative analysis between CI and satisfaction, so we used only dementia for univariate and multivariate analysis.

Reviewer 2 Report

Overall, the manuscript is written well, although minor language corrections should be applied. The methodology (patient selection, use of questionnaires) itself is sound. There are, however, several major concerns:

1.) The authors classified dementia by Mini Mental State Examination score<23. They do not give the exact results of the MMSE nor the areas of highest incapacity of the MMSE. 36.2% of patients were classified with dementia in this population. Due to the lack of MMSE results it is difficult to estimate whether the patients with dementia are able to valid questions regarding conveniance and satisfaction.

2.) 17% of patients received their drugs by care-givers or from a pre-sorted pill-box. One wonders how these patients would interprete or describe the term "conveniance", even more as patients who administered the drugs themselves reported higher conveniance.

3.) The intake of OACs is (normally) without any effect to the patient, while for example antihypertensive drugs might induce dizziness. Patients normally do not "feel" that they take OACs. Thus, on wonders whether the results indeed demonstrate conveniance and satisfaction of taking OACs, or rather, of taking any drug(s) at all. This being said, either a control group taking the same amount of drugs but no OAC is missing, or the patients should bea sked for the conveniance and satisfaction of taking any drug at all, and of taking each of the other drugs of this particular patient.

Author Response

Dear Editoriar Board,

Dear Reviewer,

We would like to thank the Editorial Board and the Reviewers for their feedback and constructive recommendations regarding our original research paper entitled:  “The influence of frailty syndrome and dementia on the convenience and satisfaction with oral anticoagulation treatment in elderly patients with atrial fibrillation.”

We would like to respond to this opinion based on our careful revision.

Sincerelly yours,

Katarzyna Lomper, corresponding author

Reviewer2

Overall, the manuscript is written well, although minor language corrections should be applied. The methodology (patient selection, use of questionnaires) itself is sound. There are, however, several major concerns:

The authors classified dementia by Mini Mental State Examination score<23. They do not give the exact results of the MMSE nor the areas of highest incapacity of the MMSE. 36.2% of patients were classified with dementia in this population. Due to the lack of MMSE results it is difficult to estimate whether the patients with dementia are able to valid questions regarding conveniance and satisfaction.

We would like to the Reviewer 2 for insightful analysis of our paper and for valuable suggestions on the submitted manuscript. In our study, 36.2% of patients had cognitive impairment with dementia, with a mean MMSE questionnaire score of 18.89 points which is equivalent to the presence of mild dementia (19-23 points). We want to emphasize that in our study we used the Mini Mental State Examination screening tool to assess cognitive function, and all questionnaires were completed in the presence of the investigator. We added information about the mean score for dementia in the paper (table number 1)

17% of patients received their drugs by care-givers or from a pre-sorted pill-box. One wonders how these patients would interprete or describe the term "conveniance", even more as patients who administered the drugs themselves reported higher conveniance.

In our study, patients taking OAC medications alone reported higher values for both satisfaction and convenience during OAC treatment, although questions about convenience focus not only on issues related to whether the patient has difficulty taking pills or adjusting medication doses, but also on recommendations for dietary restrictions, taking OAC treatment, e.g., during trips and travel, follow-up visits, and dependence on other family members. However, we want to thank very much for the valuable tip, which we will certainly use in our next paper.

The intake of OACs is (normally) without any effect to the patient, while for example antihypertensive drugs might induce dizziness. Patients normally do not "feel" that they take OACs. Thus, on wonders whether the results indeed demonstrate conveniance and satisfaction of taking OACs, or rather, of taking any drug(s) at all. This being said, either a control group taking the same amount of drugs but no OAC is missing, or the patients should bea sked for the conveniance and satisfaction of taking any drug at all, and of taking each of the other drugs of this particular patient.

Patients were informed during their participation in the study that the questions of the satisfaction and convenience questionnaire (PACTQ-2) pertain only to the anticoagulant treatment received. We want to thank for this valuable suggestion. We would like to point out that in our study, patients treated with VKA received lower values of convenience. In the discussion section, we added information that: In clinical practice, VKA and NOAC are used in anticoagulation treatment. Although we expected higher values of convenience and satisfaction in the group of patients taking NOACs, in our study we have not shown statistically significant differences depending on the use of the OAC treatment (VKA or NOAC). However, patients taking NOAC received higher convenience values.

Round 2

Reviewer 2 Report

During revision, the authors added information about the mean score of the results of the MMSE in the patients with dementia. This allows estimating the grade of mental deterioration. The other concerns were not addressed sufficiently, especially not the questions whether the answers of the participants are not related to the intake of oral anticoagulants only, but rather to the intake of medication at all. For answering this, a control group or questions regarding convenciance or satisfaction of overall drug intake had been necessary. To answer this issue by the item itself - intake of NOAK vs. VKA - is not sufficient to my regard.

However, apart from this concern the manuscript is suitable for publication.

Author Response

We want to thank the Reviewer for his insightful analysis of our paper. The Perception of Anticoagulant Therapy Questionnaire, PACTQ (Prins MH, Marrel A, Carita P, Anderson D, Bousser MG, Crijns H, Consoli S, Arnould B. Multinational development of
a questionnaire assessing patient satisfaction with anticoagulant treatment: the 'Perception of Anticoagulant Treatment Questionnaire' (PACT-Q). Health Qual Life Outcomes. 2009 6;7:9)
is dedicated to patients receiving anticoagulant treatment and its questions relate to the anticoagulant treatment. A control group study would have to be conducted among patients receiving anticoagulant treatment for indications other than atrial fibrillation e.g. deep vein thrombosis.  We believe that performing the analysis indicated by the reviewer in future work will be an interesting observation, so we will try to use this valuable hint in the future.

This manuscript is a resubmission of an earlier submission. The following is a list of the peer review reports and author responses from that submission.

Round 1

Reviewer 1 Report

This study attempted examining impacts of frailty syndrome and dementia on the oral anticoagulation treatment convenience and satisfaction among elderly patient with atrial fibrillation. The study is of cross-sectional and observational based on a survey of small sample of patients from one hospital.

Many concerns over the study are listed below.

Title

  1. It was too long.
  2. It was not clear that the word “its” in the “correlation with quality of life” was with the influence or with satisfaction.
  3. The “elderly atrial fibrillation population” seemed too grand for a small sample of local hospital, which could misled to a representative study that was not in this case.

Abstract

  1. The study sample was not defined.
  2. There was inconsistency in words such as PACT-Q2 in one place, and PACTQ-2 in another.
  3. Readers could get lost in the abbreviation such as EHRA class III/IV without details.
  4. Reporting results from single factor analysis could be misleading.

Introduction

Authors made it understood that atrial fibrillation (AF) increases the risk of stroke and stroke can predict cognitive impairment (CI), and effective treatment for AF could reduce the risk of stroke as well as CI, if AF patients have good adherence to treatment. However, I was lost in the rest of introduction and had some questions in mind.

  1. Both stroke and CI could be consequences (dependent variables in statistical terms) of inadequate treatment of AF (independent variable), why were they be treated as independent variables and AF treatment the dependent variable, a reversed relationship?
  2. What could be the clinical implication of the study on relationship between a treatment perceived by patients and possible consequences such as stroke or CI? Does it mean the treatment should be given to AF patients who were yet to have stroke or CI for better adherence, and hence better clinical effects?
  3. It is possible that CI patients did not have good adherence to most of oral treatment for many illness, why was it a problem here for AF oral treatment?
  4. Is it possible that an AF patient had both stroke and CI, instead of the one or the other? How was this situation relavent to the study?

Material and methods

  1. Were participants out patients or inpatients?
  2. What were reliability of the study tools?
  3. Who did data collection? For self-reporting questionnaires, who helped patients with CI or Dementia filled the forms? What procedures in place to ensure quality and validity of data returned from patients?

Statistical analysis

  1. What was the purpose of uni-variate analysis?
  2. The study had 23 variables on social-demographic and clinical characteristics of 116 participants. For multivariate regression analysis following a rule of thumb, only about ten variables could be entered in the model at same time due to small sample size to avoid over fitting. What method did author use to deal with this problem or ignored it completely?

Results

  1. Some clarity and corrections were required on results
  • Table 1, how comorbidities were defined?
  • Table 2, inconsistency on the presentation of P value, i.e. P followed by a value VS by p= a value.
  • Table 3, the word ‘jednoc2ynnikowa’ under Single-factor should be removed.

  1. According what in tables 3 and 4, too many variables were included in the regression analysis which would reduce statistical power of testing and result model over fitting. Therefore results based on the model should not be reliable nor robust.
  2. Correlation analysis between convenience/satisfaction and Qol did not mean much, which completely ignored the strong relationship pathway between stroke/CI and Qol.

Discussion

A major issue was that results from single-factor analysis should only be a reference in the discussion, and the results from multivariate analysis was the focus for discussion. Due to the inadequate using of regression analysis, possible overfitting and low statistical power lead to a flawed results of the study.

Conclusions

The clinical implication of the study was not clear.

Reviewer 2 Report

This paper was very well written describing a well designed and conducted study about an important topic such as anticoagulation therapy in the framework of atrial fibrillation.

The authors presented in Table 2 a comparative analysis to study the relationship between socio-demographic and clinical variables and the convenience and satisfaction of the patients included in the study. In the analysis shown in Table 3 some of these variables were not taken into consideration, for example alcohol use. It seems some variables were excluded from the second analysis, I suggest to the authors to add to the paper some sentences to explain the methods used to select the variables used in the Table 3 analysis.

The authors used the linear regression to test the association under analysis. I suggest the authors to add few sentences about their verifications of the statistical assumptions of the linear regression model, for example the Gaussian distribution.

Please update:

Table 1: “% of the total grupygrupy”

Table 2: “Frailty symdromme”

Table 3: “Single-factor jednoczynnikowa” , please remove the polish word

Table 3: “Permnament AF”